# Achievements, Developments and Future Challenges in the Field of Bioherbicides for Weed Control: A Global Review

**DOI:** 10.3390/plants11172242

**Published:** 2022-08-29

**Authors:** Jason Roberts, Singarayer Florentine, W. G. Dilantha Fernando, Kushan U. Tennakoon

**Affiliations:** 1Future Regions Research Centre, Institute of Innovation, Science and Sustainability, Federation University Australia, Ballarat 3350, VIC, Australia; 2Department of Plant Science, University of Manitoba, Winnipeg, MB R3T 2N2, Canada; 3Future Regions Research Centre, Institute of Innovation, Science and Sustainability, Federation University Australia, Berwick Campus, Berwick 3806, VIC, Australia

**Keywords:** agriculture, herbicides, land management, mycoherbicides, sustainability

## Abstract

The intrusion of weeds into fertile areas has resulted in significant global economic and environmental impacts on agricultural production systems and native ecosystems, hence without ongoing and repeated management actions, the maintenance or restoration of these systems will become increasingly challenging. The establishment of herbicide resistance in many species and unwanted pollution caused by synthetic herbicides has ushered in the need for alternative, eco-friendly sustainable management strategies, such as the use of bioherbicides. Of the array of bioherbicides currently available, the most successful products appear to be sourced from fungi (mycoherbicides), with at least 16 products being developed for commercial use globally. Over the last few decades, bioherbicides sourced from bacteria and plant extracts (such as allelochemicals and essential oils), together with viruses, have also shown marked success in controlling various weeds. Despite this encouraging trend, ongoing research is still required for these compounds to be economically viable and successful in the long term. It is apparent that more focused research is required for (i) the improvement of the commercialisation processes, including the cost-effectiveness and scale of production of these materials; (ii) the discovery of new production sources, such as bacteria, fungi, plants or viruses and (iii) the understanding of the environmental influence on the efficacy of these compounds, such as atmospheric CO_2_, humidity, soil water stress, temperature and UV radiation.

## 1. Introduction

Agricultural and environmental weeds cause significant economic and environmental damage to many ecosystems globally as they compete against, and displace, commercially important products and native and pastoral species [1,2,3,4,5]. It is estimated that the annual economic impact related to the management costs and production losses to the agricultural industry caused by these weeds is over AUD 3.3 billion in Australia [1], USD 11 billion in India [3] and USD 26 billion in the United States of America [4]. Exacerbating this issue is the fact that, over several decades, the primary method to control these weeds has been with synthetic herbicides, which accounted for over 44% of all pesticides sold around the world [5,6,7,8]. Although these compounds have shown immediate success in controlling numerous weeds, their long-term and repeated application can have severe ecological consequences [9,10]. Not only do they heavily pollute the environment, but their repeated application has also been linked to the development of herbicide resistance to several modes of action in several weed species of concern [11,12]. To date, there have been reports of more than 500 cases of herbicide resistance in 260 weed species, which involve 167 herbicides and 23 modes of action across 70 countries [12]. In order to help overcome the development of herbicide resistance, new systems with different modes of action are urgently required. However, notwithstanding the significant contributions to research in this field, there has been little development and success in this area over the past three decades [5,10]. In this regard, alternative, eco-friendly and economically viable weed management strategies that target different aspects of a plant metabolism are urgently required [13]. 

One alternative strategy that has shown considerable interest by both land managers and researchers is the use of bioherbicides [9,13,14,15,16,17]. A bioherbicide is defined as a phytopathogenic microorganism or a microbial phytotoxin that can be applied to a plant to reduce its vigour or cause its death [18]. These compounds are sourced from living organisms and contain specialised allelochemicals, genetic material or plant extracts that have been engineered or manipulated in order to overcome targeted plant defence systems [18,19,20]. To be successfully integrated within weed management programs, these bioherbicides need to (i) have a specialised and suitable formulation, (ii) be economically sustainable, (iii) cause a high mortality rate on the targeted plant and (iv) have very limited or no impact on the surrounding natural environment and human health [20,21]. Consequently, this review will explore the global literature to assess the suitability of the developments of bioherbicides sourced from bacteria, fungi, plants or viruses. It will also identify research challenges and areas that still require further research for their long-term successful field application in a sustainable manner. 

## 2. Bioherbicides and Their Mechanisms

Unlike classical biological control, bioherbicides use formulations of plant pathogens that are manipulated to produce large amounts of infectious material [21]. This material is commonly inoculated into the target organism via a liquid spray or solid granule applied directly to the body of the plant [22], which must then infiltrate its system for it to be effective [9,23]. Research has shown that once a bioherbicide enters a plant, it begins to produce several enzymes, including amylases, cellulases, ligninases, pectinases, peptidases, phospholipases or proteases, each of which assist in the degradation of cell walls, lipid membranes and proteins [24]. This breakdown allows any pathogens within the bioherbicide to further spread and gain easier access throughout the weed [24,25]. It has also been reported that phytotoxic secondary metabolites and peptides can contribute to weed control by modifying gene expression and interfering with plant metabolism and defence mechanisms [26]. Inoculated bioherbicides can cause several metabolic changes to a targeted organism such as (i) reducing the function of cellular activities, enzymes and hormones, which can lead to decreased nutrient absorption, (ii) deregulation in photosynthesis and membrane permeability, (iii) induced lipid peroxidation and (iv) inhibition of seed germination and development [24,27,28]. Reduced nutrient uptake can impact chloroplast development and cause chlorosis, whilst changes in plant hormones can result in phenolic compounds which inhibit the gibberellin pathway, increasing the accumulation of abscisic acid, jasmonic acid and salicylic acid [27,28]. This accumulation of compounds can alter a plant’s photosynthetic rates, increase oxidative stress and influence stomata closure, all of which reduce plant growth and increase senescence [27]. Although this action is common for many bioherbicides, recent research has shown that different pathogens can influence a targeted plant in contrasting ways [23]. 

## 3. Bacteria

Whilst the development of bacteria-based bioherbicides has shown encouraging results in the control of various weeds (Table 1), one of the largest barriers and challenges in their development is identifying and sourcing suitable material that directly interacts with and controls a targeted weed [20]. Recent achievements and developments in this field have identified several suitable bacteria that are capable of suppressing plant growth (Table 1), and of these recognized sources, *Xanthomonas campestris* pv. poae (JT-P482 strain) and *X. campestris* (LVA-987 strain) have shown the greatest success in controlling the growth of various turf grass weeds (Table 1) [29,30,31]. Since their identification, these strains have been formulated as a bioherbicide in Japan under the name Camperico™ [29,31]. However, despite some success in controlling selected weeds (Table 1), further research has suggested that this bioherbicide is strongly influenced by environmental conditions, and subsequently has been shown to require a dew period of 25 °C to sustain over 60% mortality rate [31]. In this regard, it would be of value for future research to consider evaluating the different environmental conditions, such as climate, humidity, and temperature, that may influence the success of selected bioherbicides [31]. This information would be of singular value to land managers by ensuring that they can obtain the highest success rate whilst using these products. 

Several other sources of bacteria, including *Curtobacterium* sp. (MA01) [38], *Pseudomonas fluorescens* (D7 strain) [33,34], *P. fluorescens* (WH6 strain) [39,40], *P. fluorescens* (BRG100 strain) [22,36] and *P. viridiflava* (CDRT_C_14 strain) [37] have also shown promising signs for suppressing various weeds and thus could be used as a bioherbicide (Table 1). Despite showing success, it is important to note here that many bacteria-based bioherbicides may take longer to suppress a weed compared to synthetic herbicides [34]. For example, it has been determined that *P. fluorescens* may take five to seven years to completely suppress an infested area, an approach which can be both time consuming and costly. It has also adversely affected the surrounding native grasses [34,39]; thus, these findings indicate that there is limited potential for it as a suitable bioherbicide unless problems with its economic value, action time and environmental impact can be addressed. 

It would also be valuable to further investigate the integrated use of bacteria-based bioherbicides with other management actions such as fire management, herbivory or manual control. However, although this might seem like a promising approach, combined management actions with bioherbicides may not always provide adequate control of a weed. In this respect, evidence by Pyke et al. [35] has shown that *Bromus tectorum* L. (cheatgrass) was unsuccessfully controlled with a bacteria-based bioherbicide (sourced from *P. fluorescens*) alongside native sowing after fire. Despite this complexity, future research in this field is still urged as different bioherbicides and weed combinations may interact differently. It is also commonly known that many integrated weed management programs provide greater confidence in the long-term, sustainable management of weeds [41]. 

## 4. Fungi (Mycoherbicides)

The development of fungi-based bioherbicides has shown increasing success for the control of various weeds (Table 2) [6,9,42,43,44,45,46], and achievements in this field have dated back to the 1950s when Russian scientists mass-produced and formulated the spores of *Alternaria cuscutacidae* to control holoparasite *Cuscuta* species (dodder) [18]. Since then, multiple mycoherbicides have been established and made commercially available in Australia, Canada, China, South Africa, the Netherlands and the USA [6,9,42,43,44,45,46,47,48,49]. Among these products, BioChon™ [43], Chontrol™/Ecoclear™ [6,50], Myco-Tech™ [6,50] and Stumpout^®^ [47,49] have been developed for the control of woody weeds (Table 2). These mycoherbicides, which are often applied via a mycelium paste to the target weed’s cut stump, eventually block the vascular system of the plant with vigorously growing mycelia, whilst preventing it re-sprouting and increasing its decomposition [6,7,43,50]. Since mechanical cutting of the plant close to ground level is required before mycoherbicides can be applied [6,7], such an approach can become time-consuming and costly when dealing with large infestations. To combat this issue, a recently developed stem-injected mycoherbicide in capsule form has been made available, which can be mechanically drilled and released into the target plant without exposing the operator to a hazardous situation [46]. Commercialised under the name Di-Bak Parkinsonia™, this bioherbicide has shown high success in controlling *Parkinsonia aculeata* L. (parkinsonia) through formulation of the fungi *Lasiodiplodia pseudotheobromae*, *Macrophomina phaseolina* and *Neoscytalidium novaehollandiae* [46]. Not only does this method limit the use of herbicides within the environment and the need to physically cut and remove the plant, but after establishment, this mycoherbicide can actually progress through a population of *P. aculeata* and prevent future recruitment from the seedbank [46]. It would thus be of value for future research to investigate a range of mycoherbicides that might be capsule stem-injected into other woody weeds, noting that the greatest challenge for this approach would be in identifying suitable mycoherbicides that are capable of acting in this fashion without impacting on adjacent native species. 

The literature suggests that fungi from the genus *Colletotrichum* are one of the most-used species within mycoherbicide formulations [48,54,56,57,58,71,72,73]. Previous research developments using this genus have resulted in the production of several mycoherbicides (Table 2) including BioMal^®^ (sourced from *C. gloeosporioides* f. sp. malvae) [56,57,74], Collego™/LockDown™ (sourced from *C. gloeosporioides* f. sp. aeschynomene) [44,56], Lubao1 and Lubao 2 (sourced from *C. gloeosporioides*) [45], Velgo^®^ (sourced from *C. coccodes*) [54] and *C. truncatum* (not yet commercially developed) [58]. Although this genus of fungi has shown success in controlling various weeds around the world, many of these products have failed to become widely available as (i) they are often more expensive than synthetic herbicides, (ii) their success is often less assured compared to synthetic herbicides and (iii) they have a small niche range [75,76]. These drawbacks have resulted in a lower demand for these products because their actions limit their use [75,76]. These limitations have also been noted in several other developed mycoherbicides such as Casst™ (sourced from *Alternaria cassiae*) [42], DeVine^®^ (sourced from *Phytophthora palmivora*) [65], Dr. Biosedge^®^ (sourced from *Puccinia canaliculata*) [67], Sarritor™ (sourced from *Sclerotinia minor*) [51,68], Smolder^®^ (sourced from *Alternaria destruens*) [51,52] and Woad Warrior^®^ (sourced from *Puccinia thalaspeos*) [7] (Table 2). In this regard, future research should aim to develop mycoherbicides that can be sustainably and economically applied for the long-term control of weeds. This could be achieved by (i) improving the cost and production of these products for large-scale use, (ii) promoting, encouraging and educating land managers on the use of mycoherbicides, (iii) reducing the danger of peripheral damage to adjacent species, and (iv) improving the efficacy of some existing products [77,78]. 

In this regard, recent developments in this field have identified several other sources of fungi that have the potential to be formulated as a mycoherbicide, but these have yet to be commercialised for application (Table 2). Several of these fungi include *Albifimbria verrucaria*, (formally *Myrothecium verrucaria*) [25,53], *Fusarium oxysporum* f. sp. [59], *Gibbago trianthemae* [61], *Phoma chenopodicola* [62], *Phoma macrostoma* Montagne 94–44B [16,63,64], *Pseudolagarobasidium acaciicola* [66], *Trichoderma koningiopsis* [69] and *Trichoderma polysporum* [70]. 

## 5. Plant Extracts (Allelochemicals and Essential Oils) 

Phytotoxins derived from plants, which are either allelochemicals or essential oils, have shown high potential and success in controlling various weeds (Table 3). These compounds have been reported to have several advantages over synthetic herbicides as they (i) are biodegradable, (ii) have diversity in their modes of action and (iii) are often safe to human health and do not affect non-target species [79]. Of particular interest are allelochemicals, which are secondary metabolites produced by a plant that induce harmful changes to the enzymes, genetics, hormones and metabolic processes when applied to another plant. These effects ultimately cause severe plant stress and gradual death [79,80,81], and experiments and developments in this field have identified several plant sources with the potential to be formulated as a bioherbicide (Table 3). These sources include allelopathic chemicals from *Canavalia ensiformis* de Candolle (jack bean) (50 g L^−1^) [82], *Cirsium setosum* L. (HL-1 isolate) [83], *Cynara cardunculus* L. (artichoke thistle) (ethanol + lyophilized leaf extracts) [84,85], *Juglans nigra* L. (black walnut) (>42.9% concentration) [86], *Lantana camara* L. (Lantana) [81], *Ocimum basilicum* L. (sweet basil) [87] and *Sorghum bicolor* L. (great millet) [88]. While all these sources have shown success in controlling various weeds (Table 3), it is important to note here that further research on their long-term and repeated use in agricultural and natural ecosystems is required. In particular, research on allelochemicals needs to consider their (i) phytotoxic activity, (ii) influence on surrounding species, (iii) chemical structure, (iv) mode of action and (v) ability to become safely and sustainably commercialised [89,90]. 

Essential oils extracted from plants have also shown success when used as a bioherbicide to control several weeds (Table 3) [91,92,93,94,95,96]. Essential oils can be extracted from a plant’s bark, flowers, fruits, leaves, roots or from the entire plant [97]. They can cause severe damage to the DNA, biochemical processes and cellular functions of the targeted vegetation, which can lead to its gradual death [97]. Research has shown that, since 2020, there have been several commercially available bioherbicides that use essential oils as a key ingredient. These products include Avenger Weed Killer^®^ (70% d-limonene), GreenMatch^®^ (55% d-limonene), GreenMatchEX^®^ (50% lemongrass oil), Weed Slayer^®^ (6% eugenol), WeedZap^®^ (45% clove oil and 45% cinnamon oil), and Bioweed™ (10% pine oil + sugar) [93]. More recently, another bioherbicide, known as Weed Lock^®^, has also been developed in Malaysia as a non-selective bioherbicide for a range of weed species [77]. This developed bioherbicide is absorbed through the foliage and causes chlorosis and withering in the targeted plant within hours after application [77]. However, although it has shown success in controlling various weeds, this product has only been marketed as a ready-to-use bioherbicide in small quantities. In this regard, this product is not economically sustainable for large-scale weed control [77], and it requires further scaling up of its formulation to allow for it to be applied on a large scale.

**Table 3 plants-11-02242-t003:** Plant-sourced bioherbicides and their impact on targeted weeds.

	Plant Source	Target Weed(s)	Effect ^a^	Mode of Action	Commercial	Reference
**Allelochemicals**	*Canavalia ensiformis* extract (50 g L^−1^)	*Commelina benghalensis* (Benghal dayflower)	****	Causes inhibition of plant growth and development	X	[82]
		*Ipomoea grandifolia* (little bell)	****			
	*Cirsium setosum* (HL-1 isolate)	*Chenopodium album* (goosefoot)	***	Creates high levels of phytotoxins that inhibit seed germination and plant growth	X	[83]
		*Galium aparine* (cleavers)	***			
		*Malva crispa* (Chinese mallow	***			
		*Polygonum lapathifolium* (pale knotweed	***			
	*Cynara cardunculus* (Ethanol + Lyophilized leaf extracts)	*Amaranthus retroflexus* (redroot pigweed), *Anagallis arvensis* (scarlet pimpernel), *Phalaris minor* (little seed canary grass), *Portulaca oleracea* (little hogweed), *Stellaria media* (chickweed), *Sylibum marianum* (milk thistle), *Trifolium incarnatum* (crimson clover)	****	Induces oxidative stress and disrupts physiological and biochemical functions within the plant cells.	X	[84,85]
	*Juglans nigra* (black walnut) extracts (>42.9% concentration)	*Convolvulus arvensis* (field bindweed)	****	Inhibits H^+^-ATPase activity decreases photosynthesis and reduces root, leaf and cotyledon production	NatureCur^®^ (USA). Limited availability.	[86]
		*Conyza bonariensis* (hairy fleabane)	****			
		*Conyza canadensis* (horseweed)	****			
		*Echinochloa crus-galli* (barnyard	****			
		*Ipomoea purpurea* (tall annual)	****			
		*Portulaca oleraceae* (common purslane)	****			
		*Solanum nigrum* (black nightshade)	****			
	*Lantana camara* cold and hot extracts	*Avena fatua* (common wild oats)	***	Allelopathic compounds (aromatic) present in the plant cause the suppression of plant growth and germination	X	[81]
		*Euphorbia helioscopia* (sun spurge)	***			
		*Phalaris minor* (little seed canarygrass)	***			
		*Rumex dentatus* (toothed dock)	***			
	*Ocimum basilicum* extracts	*Amaranthus* species	***	Inhibits germination, growth and root/shoot elongation	X	[87]
		*Portulaca* species	***			
	*Sorghum bicolor* (great millet)	*Amaranthus retroflexus* (redroot pigweed), *Ambrosia artemisiifolia* (common ragweed), *Cassia obtusifolia* (sicklepod), *Coronopus didyum* (lesser swinecress), *Cyperus rotundus* (purple nutsedge), *Phalaris minor* (littleseed canary grass), *Solanum nigrum* (black nightshade)	-	Inhibits photosynthetic apparatus by altering the uptake of solutes and water molecules.	X	[88]
**Essential Oils**	*Corymbia citriodora*, formerly *Eucalyptus citriodora* (formerly, oil (0.03% concentration)	*Avena fatua* (common wild oat)	****	Impacts chlorophyll and cellular membranes causing chlorophyll and cell disruption	X	[94]
		*Sinapis arvensis* (charlock)	****			
		*Sonchus oleraceus* (common sowthistle)	****			
	*Corymbia citriodora*, formerly *Eucalyptus citriodora* oil (0.06% concentration)	*Amaranthus viridis* (slender amaranth)	****	Inhibits seed germination and plant growth by affecting photosynthetic and respiratory metabolism.	X	[91]
		*Bidens pilosa* (blackjack)	****			
		*Leucaena leucocephala* (lead tree)	****			
		*Rumex nepalensis* (nepal dock)	****			
	*Corymbia citriodora*, formerly *Eucalyptus citriodora* oil (5.0 nL mL^−1^ concentration)	*Parthenium hysterophorus* (parthenium weed)	****	Causes rapid electrolyte leakage, which impacts membrane integrity.	X	[98]
	*Eucalyptus globulus* oil + *Syzygium aromaticcum* (Clove) oil (10% concentration)	*Chenopodium album* (goosefoot)	***	Causes rapid electrolyte leakage and cellular membrane disruption.	X	[95]
		*Melilotus indicus* (Indian sweet clover)	***			
		*Raphanus raphanistrum* (wild radish)	***			
		*Sisymbrium irio* (London rocket)	***			
	Lemon-scented *Eucalyptus citriodora* oil (0.07% concentration)	*Phalaris minor* (littleseed canary grass)	****	Impacts the photosynthetic and respiratory ability of treated plants.	X	[91]
	Manuka oil mixture *from Leptospermum scoparium* (manuka tree)	*Avena sterilis* (sterile oat)	****	Natural b-triketones inhibit the biosynthesis of tocochromanols and prenyl quinones.	X	[92,96]
		*Galium aparine* (clever)	****			
		*Lolium rigidum* (rigid ryegrass)	****			
	Pine oil (10% concentration) + sugar	*Nassella trichotoma* (Serrated tussock) other herbaceous and grassy weeds	-	Inhibits seed germination and plant growth.	Bioweed™	[93]

^a^ Effect: (-) = not applicable/available, *** = 50–75%, **** = 75–100% control/plant growth reduction. X = not commercially available.

Other essential oils that have also shown promising results when used as a bioherbicide include *Corymbia citriodora* Hooker, formerly *Eucalyptus citriodora* (lemon-scented gum) and *E. globulus* Labillardière (blue gum) [91,93,95], manuka oil, extracted from *Leptospermum scoparium* Forster (manuka tree) [92,96] and pine oil (10% concentration + sugar) [93] (Table 3). Furthermore, several compounds from the essential oils of citronella, cloves, lemongrass, oranges, pine oil, thyme and several other *Eucalyptus* species have shown possible bioherbicidal effects on many plant species [93]. Future research would need to investigate the formulation of these essential oils and identify which plant species they could target when applied as a bioherbicide. 

## 6. Viruses

Bioherbicides containing a viral pathogen have shown varying success and suitability for weed control (Table 4), and of the listed formulations suitable for this work, one of the most successful and promising sources is the Tobacco mild green mosaic virus (TMGMV) [19,99,100]. This virus has shown high success at controlling *Solanum viarum* Dunal (tropical soda apple) in Florida (USA), as it can cause necrotic local lesions and hypersensitive response in the species, leading to plant death within 20 to 50 days [19,99,100]. It is also important to note that viral infectious material contains nucleic acid (DNA or RNA) and needs to be introduced into the living cells through macroscopic or microscopic injuries [98]. In this regard, TMGMV needs to be formulated with a carborundum and organosilicon adjuvant to help it penetrate the plant, and it should also be applied via (i) an abrade-and-spray application, (ii) high-pressure sprayers (>80 psi) or (iii) a wiper application [100,101]. TMGMV can also remain infectious when combined with several synthetic herbicides, therefore in a suitable combination it may provide greater control [19]. Araujia mosaic virus (AMV) has also been reported as a potentially promising bioherbicide for the control of *Araujia hortorum* Brotero (moth plant) in New Zealand [102]. This virus is known to cause mosaic symptoms and leaf distortion which leads to plant death [102]. Although being a promising bioherbicide, AMV has been known to cause non-target damage to several species that are utilised by the valued *Danaus plexippus* L. (Monarch butterfly) as habitat [102]. In this regard, whilst AMV is not currently regarded to be a suitable bioherbicide unless it is further genetically modified to limit transmission to other species, such work is not widely accepted and can be extremely costly [102]. Tobacco rattle-like virus, Pepper mosaic virus (Óbuda Pepper Virus) and Pepino mosaic virus have also shown potential in controlling several weeds when formulated as a bioherbicide (Table 4) [103,104,105]. For their successful and confident application as a bioherbicide, more research is still required to determine their application style, host-specificity and formulation. 

## 7. Achievements, Developments and Future Challenges

Recent achievements and developments in the field of bioherbicides have resulted in over 22 different formulations being currently registered on the market for weed control (Table 5). Although many more have previously been developed, or are currently being formulated, low customer demand and the high costs currently associated with the production of those formulations appear to limit their long-term success (Table 1, Table 2 and Table 3). In 2016, it was reported that the global bioherbicide market was valued at USD 1.28 billion, and with continued development in this field, it is expected to further increase its market share to USD 4.14 billion by 2024 [77,106]. Although advancements in this field are clearly evident, there are still several challenges associated with the commercialisation process that may hinder their widespread success [77,107], and these barriers need to be addressed to ensure that all developed bioherbicides can be economically and commercially viable in the long term. 

One of the many barriers that restrict the use and success of bioherbicides is the influence of environmental conditions [2,23,111]. Factors that have been reported to strongly influence the success of a bioherbicide include humidity, soil type, temperature and UV light (which can degrade the bioherbicide), together with adequate water availability and quality [111]. These factors influence the formulation process and the performance of a bioherbicide, which can lower its efficiency when applied directly to a plant [111]. It has also been reported that increased temperatures and carbon dioxide (CO_2_) levels in the atmosphere, driven by anticipated climate change scenarios, are likely to significantly change weed population dynamics in the future [83,112]. In this regard, current weed management strategies may not provide adequate control of many weeds, with evidence showing that increased CO_2_ concentrations and temperature have resulted in more occurrences of herbicide resistance [113,114,115,116]. This strongly supports the use and need for improved weed management, a direction which includes bioherbicide applications. It is anticipated that a plethora of questions will continue to emerge relating to the overall impacts and modes of action of bioherbicides due to structural and physiological variations and evolutionary adaptations that will inevitably occur in weeds due to predicted changes in the climate [115,116]. This collectively demonstrates the value of embracing a multidisciplinary research approach to assess the efficacy of bioherbicides in the future. In this regard, it would also be of value for future research to investigate how these climatic scenarios may influence the performance of selected bioherbicides, allowing them to be used in future weed management programs with greater confidence.

Another barrier to the success of bioherbicides is the formulation and commercialisation process [20,23,104]. Because they contain a living biotic agent, the viability and stability of many bioherbicides need to be steadily maintained [20,117], requiring consideration of how these biological agents can remain active over the extended time frame needed from the development to application stage. This would require identification of the optimal storage conditions for a particular bioherbicide, noting that each case could vastly differ. In addition, the high costs associated with the formulation and commercialisation of many bioherbicides need to be addressed for their use to be sustainable over large areas [77]. This could be partly achieved with increased educational awareness of the benefits and use of bioherbicides, together with improvements made to technology, such as smart spray systems [78]. 

National and government regulations also restrict the widespread use of bioherbicides as they need to be formally registered with the Environmental Protection Agency [76,107,111]. This procedure can often be costly and prolong the developmental process because it is known to vary across different countries. In particular, the use and investment in bioherbicides in Australia, Canada and the USA are significantly higher compared to many European countries [9,107]. Thus, the restricted level of development work can be seen as partly a result of the uncertainties and consequent hesitation in using biologically active agents in an agricultural or natural setting [9]. To combat these uncertainties, it is suggested that governments and non-government agencies work together to identify areas where information is urgently needed for the successful, confident and long-term use of bioherbicides for weed control [2].

## 8. Conclusions

This global review of the literature has indicated that using bioherbicides for weed control can have numerous advantages over traditional synthetic herbicide applications, identifying that bioherbicides can (i) be used on herbicide-resistant weeds, (although future research should consider the development of potential bio-herbicide resistance in weeds from these bioherbicides over time), (ii) have high host-specificity in selected habitats, and (iii) be more environmentally friendly and less toxic than synthetic herbicides. However, despite showing promising signs as an emerging weed control technique, there is still a need for ongoing research to focus on highly complex processes in a concerted manner by: (i) improving the commercialisation process, (ii) finding more suitable sources from bacteria, fungi, plant extracts or viruses (in order to develop new products), (iii) understanding the different modes of action for each class of bioherbicide and (iv) to determine how environmental conditions will influence their success.

## Figures and Tables

**Table 1 plants-11-02242-t001:** Bacterial bioherbicides and their impacts on targeted weeds.

Bacterial Source	Target Weed(s)	Effect ^a^	Mode of Action	Commercial	Reference
*Curtobacterium* sp. MA01	*Petunia* spp.	-	Alters enzymatic and metabolic reactions including the degradation of protein synthesis and lipid peroxidation.	X	[32]
*Pseudomonas fluorescens* D7	*Aegilops cylindrica* (jointed goatgrass); *Bromus tectorum* (downy brome); *Taeniatherum caput-medusae* (medusa-head)	**	Colonizes root structures and interferes with the enzymes that use pyridoxal phosphate as a cofactor.	X	[33,34]
*Pseudomonas fluorescens* D7	*Bromus tectorum* (cheatgrass)	*		X	[35]
*Pseudomonas* fluorescens BRG100	*Setaria viridis* (green foxtail)	-	Interferes with plant hormones and metabolism, inhibiting roots and shoots.	X	[22,36]
*Pseudomonas viridiflava* CDRT_C_14	*Lepidium draba* (hoary cress)	-	Alters plant hormones and metabolism.	X	[37]
*Xanthomonas campestris* pv. poae (JT-P482)	*Poa annua* (annual bluegrass); *Poa attenuata* (meadow-grass)	****	Suppresses growth and causes black rot disease.	Camperico™	[29,32]
*Xanthomonas campestris* (LVA-987)	*Ambrosia artemisifolia* (common ragweed)	***	Suppresses growth and causes black rot disease.	X	[30,31,32]
	*Ambrosia trifida* (giant ragweed)	***			
	*Conyza canadensis* (marestail)	****			
	*Xanthomonas spp.* (common cocklebur)	****			

^a^ Effect: (-) = not applicable/available, * = 0–25%, ** = 25–50%, *** = 50–75%, **** = 75–100% control/plant growth reduction. X = not commercially available.

**Table 2 plants-11-02242-t002:** Fungal bioherbicides and their impacts on targeted weeds.

Fungal Source	Target Weed(s)	Effect ^a^	Mode of Action	Commercial	Reference
*Alternaria cassiae*	*Cassia obtusifolia* (sicklepod), *Cassia occidentalis* (coffee senna), *Crotalaria spectablis* (showy crotalaria)	-	Causes parasitic leaf blight and damage to the plant.	Casst™ (USA). No longer available.	[42]
*Alternaria destruens*	*Cuscuta* spp. (dodder)	-	Inhibits plant growth and development.	Smolder^®^ (USA). No longer available.	[51,52]
*Albifimbria verrucaria*, formally *Myrothecium verrucaria*	*Pueraria lobata* (kudzu)	****	Inhibits seed germination and early plant growth.	X	[25,53]
*Chondrostereum purpureum*	*Prunus serotina* (black/wild cherry)	-	Prevents stumps from resprouting and increases woody decay.	BioChon™ (The Netherlands). No longer available.	[43]
*Chondrostereum purpureum*	hardwoods and deciduous trees and shrubs	-	Causes stump decay and prevents resprouting.	Chontrol™/EcoClear™/ MycoTech™	[6,50]
*Colletotrichum coccodes*	*Abutilon theophrasti* Medicus (velvetleaf)	-	Causes inoculation damage and prevents plant growth and production.	Velgo^®^ (Canada). No longer available.	[48,54]
*Colletotrichum gloeosporioides*	*Echinochloa crus-galli* (barnyard grass)	-	Causes severe infection and leaf spot disease in the plant.	Lubao 1 and Lubao 2 (China). Limited availability.	[45,55]
	*Cuscuta* chinensis Lamarck (Chinese dodder) and *Cuscuta australis* r brown (Australian dodder)	****			
*Colletotrichum gloeosporioides* f. sp. aeschynomene	*Aeschynomene virginica* (jointvetch)	****	Induces anthracnose lesions on the plants’ stems.	Collego™/LockDown™	[7,9,44,56]
	*Aeschynomene indica* (Indian jointvetch)	****			
	*Sesbania exaltata* (hemp sebania)	****			
*Colletotrichum* gloeosporioides f. sp. malvae	*Malva pusilla* (round-leaved mallow)	-	Causes lesions within the plant’s flowers, leaves and stems.	BioMal^®^ (Canada) No longer available.	[57]
*Colletotrichum truncatum*	*Bidens pilosa* (beggartick)	****	Inhibits plant growth and seed germination.	X	[58]
*Cylindrobasidium laeve*	*Acacia mearnsii* (black wattle), Acacia *pycnantha* (golden wattle), *Poa annua* (winter grass)	-	Accelerates the decomposition of stumps and roots.	Stumpout™	[47,49]
*Fusarium oxysporum* f. sp. *orthoceras*	*Orobanche* spp. (broomrape)	***	Causes lesions on the leaves.	X	[59]
*Fusarium fujikuroi* Sawada.	*Cucumis sativus* L. (cucumber) and *Sorghum bicolour* L. (great millet)	**	Causes chlorosis and necrosis.	X	[60]
*Gibbago trianthemae*	*Trianthema portulacastrum* (horse purslane)	-	Causes stem blight and leaf spot disease	X	[61]
*Lasiodiplodia pseudotheobromae*, *Macrophomina phaseolina* and *Neoscytalidium novaehollandiae*	*Parkinsonia aculeata* (parkinsonia)	-	Produces harmful toxins and enzymes that disarm the plants’ defence mechanisms, leading to cell and tissue degradation.	Di-Bak Parkinsonia™	[46]
*Phoma chenopodicola*	*Chenopodium album* (lamb’s quarter)	-	Causes extensive necrotic lesions	X	[62]
*Phoma macrostoma* Montagne 94–44B	Broadleaf weeds such as *Taraxacum officinale* (dandelion)	-	Colonizes and passes into the root system which causes mycelium to obstruct nutrient uptake.	Phoma^®^	[16,63,64]
*Phytophthora palmivora*	*Morrenia odorata* (milkweed vine)	-	Causes a root infection in the plant which leads to its death.	DeVine^®^ (USA). No longer available.	[9,65]
*Pseudolagarobasidium acaciicola*	*Acacia cyclops* (coastal wattle)	****	Causes seed mortality and plant death.	X	[66]
*Puccinia canaliculata*	*Cyperus esculentus* (yellow nutsedge)	-	Inhibits the reproductive process and seed germination in the species	Dr. Biosedge^®^ (USA). No longer available.	[67]
*Puccinia thalaspeos*	*Isatis tinctoria* (dyer’s woad)	-	Infects first-year plants and impacts flowering and seed formation in the following year.	Woad Warrior^®^	[7]
*Sclerotinia minor*	*Araxacum officeinale* (dandelion), broadleaf	-	Absorbs plant tissue.	Sarritor™	[51,68]
*Trichoderma koningiopsis*	*Euphorbia heterophylla* (Mexican fire plant)	***	Increases enzymatic material (cellulase and lipase) which causes increased damage to the plant.	X	[69]
*Trichoderma polysporum* (Louk: Fr.) Rifai.	*Avena fatua* (common wild oats)	****	Produces several secondary metabolites that have antifungal activities and prevent plant growth and germination.	X	[70]
	*Chenopodium album* (goosefoot)	****			
	*Elsholtzia densa* (dense Himalayan mint)	****			
	*Lepyrodiclis holosteoides* (false jagged chickweed)	****			
	*Polygonum aviculare* (common knotgrass)				
	*Polygonum lapathifolium* (pale persicaria)				

^a^ Effect: (-) = not applicable/available,** = 25–50%, *** = 50–75%, **** = 75–100% control/plant growth reduction. X = not commercially available.

**Table 4 plants-11-02242-t004:** Viral bioherbicides and their impact on targeted weeds.

Virus Source	Target Weed(s)	Effect ^a^	Mode of Action	Commercial	Reference
Araujia Mosaic Virus	*Araujia hortorum* (moth plant)	-	Causes mosaic symptoms and leaf distortion in the plant.	X	[102]
Pepper mosaic virus (Óbuda Pepper Virus)	*Solanum nigrum* (black nightshade)	-	Causes biomass reduction and increased seed dormancy.	X	[104]
Tobacco rattle-like virus	*Impatiens glandulifera* (Himalayan balsam)	-	Develops necrotic spots on the plant.	X	[105]
Tobacco mild green mosaic virus	*Solanum viarum* (tropical soda apple)	****	Triggers a hypersensitive response in *S. viarum* and causes necrotic local lesions.	SolviNix™ LC and WP (liquid concentrate and wettable powder)	[19,99,100]

^a^ Effect: (-) = not applicable/available, **** = 75–100% control/plant growth reduction. X = not commercially available.

**Table 5 plants-11-02242-t005:** Currently available bioherbicides on the market for weed control around the world. Information adapted and sourced from: [6,7,9,11,29,32,46,51,93,100,108,109,110].

Commercial Name	Active Constituents	Use/Target Plant(s)	Country Available	Released
Avenger Organic Weed Killer^®^	d-Limonene and castor oil	Grass and broadleaf weeds	USA	N/A
Barrier H°	22.9% citronella oil	Ragwort	Europe, Japan, USA	2015
Beloukha^®^/Scythe^®^	Rapeseed oil, nonanoic acid and pelargonic acid	Non-selective control of seedlings and young weeds	Australia, USA	N/A
Bialaphos^®^	*Streptomyces hygroscopicus*	Broad-spectrum and post-emergence bioherbicide	Eastern Asia	2016
Bioweed™	Pine oil (10% concentration) + sugar	Herbaceous and grassy weeds	Australia	N/A
Camperico™	*Xanthomonas campestris pv. poae* (JT-P482)	Turf grass weeds	Japan	1997
Di-Bak Parkinsonia™	*Lasiodiplodia pseudotheobromae*, *Macrophomina phaseolina* and *Neoscytalidium novaehollandiae*	*Parkinsonia aculeata* (parkinsonia)	Australia	2013
GreenMatch^®^	Lemon grass oil	Broadleaf and grassy weeds	USA	2008
Katana^®^	Pelargonic acid	Broadleaf and grassy weeds	USA	2016
Lockdown^®^/Collego™	Flumioxazin and *Colletotrichum gloeosporioides* f. sp. aeschynomene	Residual control of various broadleaf weeds	USA	N/A
Matratec^®^	Clove oil, lactic acid, lecithin, n-butyl ester and wintergreen oil	Broad-spectrum, non-selective	USA	N/A
Myco-Tech^®^/Chontrol^®^/EcoClear™	Acetic acid, citric acid and *Chondrostereum purpureum*	Non-selective to green foliage and deciduous trees and shrubs	Belgium, Canada, The Netherlands	2005
Opportune™	*Streptomyces* strain RL-110 T	Pre/Post emergent herbicide (broadleaf and sedges)	USA	2013
Organic Interceptor^®^	Pine oil	Knockdown and pre-emergent herbicide	New Zealand	N/A
Organo-Sol^®^/Kona™/Bioprotec™	Lactic acid, citric acid, *Lactobacillus rhamnosus* (LPT–21), *L. casei* (LPT–111), *L. lactis* ssp. *cremoris* (M11/CSL), *L. lactis* ssp. *Lactis* (LL64/CSL and LL102/CSL)	Non-selective, post-emergent herbicide	Canada	2010
Phoma^®^	*Phoma macrostoma* 94–44B (Macrocidins A, B)	Broad spectrum of broadleaf weeds	Canada, USA	2016
Sarritor^®^	Flumetsulam and *Sclerotina minor*	Broadleaf weeds	Australia, Canada	2007
SolviNix™ LC and WP	Tobacco soft green mosaic, Tobamovirus cepa U2	Tropical soda apple (*Solanum viarum*)	USA	N/A
Stump out™	Sodium bicarbonate and *Cylindrobasidium laeve*	*Acacia* and *Poa* species	South Africa	1997
Weed Slayer^®^	Eugenol, clove oil, molasses	Grassy weeds	USA	N/A
WeedZap^®^	Cinnamon oil, clove oil, lactose and water	Non-selective, small broadleaf and grassy weeds	USA	N/A
Woad Warrior^®^	*Puccinia thalaspeos*	*Isatis tinctoria* (dyer’s woad)	USA	2002

## Data Availability

Not applicable.

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
