# Peer review of "Achievements, Developments and Future Challenges in the Field of Bioherbicides for Weed Control: A Global Review"

_plants, 2022, doi:10.3390/plants11172242_

Round 1
Reviewer 1 Report
Overall, this is a good paper summerizing bioherbicides. I quite enjoy reading it. Good job!
Reviewer 2 Report
The Authors have considered a scientific argument of great relevance. The weed control by non-hazardous herbicides for the environment ad human health is a complex worldwidw challenge in agriculture with considerable economic repurcussions
In their review the Authors have well introduced the issue by a careful and in depth literature analysis (they reported and considered many and appropriate references)
They analyzed the pros and cons of the use of bioherbicides of different natural origin critically and extensively
They reported also the action mechanism of the different bioherbicide taken in consideration the action mechanism (important for the biologists and biotechnologists audience) and administration modalities
In the conclusion they highlight the necessary steps in this field of research with an unavoidable reference to economic considerations and the strict need to acquire an increasing awareness about the use of biological strategies in the containment of weeds
The paper well summarizes the state of the art of this field of research and well introduces important questions to those who approach this topic
In my opinion this paper deserves the pubblication
